# Measurements of air-sea gas transfer velocities in the Baltic Sea

Nagel, Leila[a], Krall, Kerstin E.[1], and Jähne, Bernd[1,2]

[1]Institute of Environmental Physics, Heidelberg University, Im Neuenheimer Feld 229, 69120 Heidelberg, Germany
[2]Heidelberg Collaboratory for Image Processing, Heidelberg University, Berliner Straße 43, 69120 Heidelberg, Germany
[a]previously at: Institute of Environmental Physics, Heidelberg University, Im Neuenheimer Feld 229, 69120 Heidelberg, Germany

**Correspondence:** K. E. Krall
(Kerstin.Krall@iup.uni-heidelberg.de)

**Abstract.** Heat transfer velocities measured during three different campaigns in the Baltic Sea using the Active Controlled Flux Technique (ACFT) with wind speeds ranging from 5.3 to 14.8 m s$^{-1}$ are presented. Careful scaling of the heat transfer velocities to gas transfer velocities using Schmidt number exponents measured in a laboratory study allows to compare the measured transfer velocities to existing gas transfer velocity parameterizations, which use wind speed as the controlling parameter. The measured data and other field data clearly show that some gas transfer velocities are much lower than those based on the empirical wind speed parametrizations. This indicates that the dependencies of the transfer velocity on the fetch, i. e., the history of the wind and the age of the wind wave field, and the effects of surface active material need to be taken into account.

## 1   Introduction

The transfer of a trace gas across the air sea interface is commonly characterized by the gas transfer velocity $k$, which links the gas flux $j$ with the concentration difference across the interface, $\Delta c$,

$$j = k\Delta c. \tag{1}$$

Traditionally, $k$ is parameterized with the wind speed measured in 10 m height, $u_{10}$, since wind speed is the most readily available parameter. Different authors proposed different functional dependencies between $k$ and $u_{10}$, for example a gradual transition from a smooth to a wavy regime (Jähne, 1982), piecewise linear (Liss and Merlivat, 1986), linear and quadratic terms (Nightingale et al., 2000), quadratic (Wanninkhof, 1992) or cubic (Wanninkhof and McGillis, 1999).

   Wanninkhof et al. (2009) gives an overview of the most commonly used techniques to measure the gas transfer velocity. In the last decades, the dual-tracer technique, especially with the tracer pair $^3$He/SF$_6$, as well as eddy covariance measurements of the gases CO$_2$ and dimethylsulfide (DMS) have become state of the art measuring the gas transfer velocity in situ. A recent review article by Ho et al. (2011) proposed

$$k_{600} \, [\mathrm{cm\,h}^{-1}] = 0.262 \pm 0.022 u_{10}^2 \, [u_{10} \, \mathrm{in\,m\,s}^{-1}] \tag{2}$$

as the best fit to all available $^3$He/SF$_6$ dual tracer data points, where $k_{600}$ denotes the transfer velocity scaled to a CO$_2$-equivalent transfer velocity at 20$^o$C. However, mass balance techniques such as the dual tracer method have a large time

constant of up to weeks and large spatial scales of a few tens of kilometers, smoothing away varying micrometeorological and surface conditions (e. g. the degree of surface contamination by surface active material).

In contrast, the eddy covariance method provides measurements of the gas transfer velocity on time scales below 1h and spatial scales of a few kilometers. However, bin averaging over wind speed intervals is frequently necessary, since even under idealized conditions, not all realizations of the turbulent field can be measured, so that each single flux measurement obtained during a 30 min time period is still uncertain (Garbe et al., 2014).

In this study, the active controlled flux technique (ACFT), a thermographic technique, is used, which is capable of measuring the heat transfer velocity with a temporal resolution of about 20 minutes, which can then be scaled to gas transfer velocities. This technique is described in Sect. 3.1. The ACFT was deployed during three cruises in the Baltic Sea to investigate the variability of the transfer velocities under field conditions.

Earlier measurements of the gas transfer velocity in the Baltic sea are sparse. Weiss et al. (2007) used the eddy covariance technique to measure the transfer of $CO_2$ in the Arkona Basin, and Rutgersson et al. (2008) used the same technique in the Gotland sea. Both studies found a very high variability of the gas transfer velocity.

## 2  Factors influencing air-sea gas exchange

The common approach is to parameterize the gas transfer velocity with wind speed alone. However, a wealth of studies have shown that a multitude of factors influence gas transfer, for example the contamination of the water surface with surface active material (e.g. Frew et al. (2004); Salter et al. (2011)), bubble entrainment (e.g. Woolf et al. (2007); Crosswell (2015)), fetch (e.g. Zhao et al. (2003); Woolf (2005)), rain (e.g. Zappa et al. (2009); Harrison et al. (2012)) and convective mixing (e.g. Rutgersson et al. (2011)).

Since the method discussed in this paper is insensitive to bubble contributions and can only be used to measure the interfacial part of the air sea gas transfer, and no measurements were performed in rain conditions, only the influence of surface active material and fetch will be discussed here.

### 2.1  Surfactants

One factor contributing to the disagreement between gas transfer velocities measured at the same wind speed even with the same measuring technique are surface active materials (surfactants), which reduce the gas transfer velocity. This reduction in the gas transfer velocity in the presence of surfactants is not caused by the additional diffusion of the gas through the monomolecular surfactant layer at the water surface (Frew et al., 1990), but by hydrodynamic effects in the mass boundary layer. Surfactant presence at the water surface inhibits eddy motion close to the surface and reduces fluid velocities. Upwelling at the surface is hindered by a reduction in the surface divergence due to the visco-elastic properties of the surfactant (McKenna and Bock, 2006). Vertical velocity fluctuations near the interface are considered vital to gas-transfer enhancement. Decreased vertical transport of fresh fluid towards the water surface results in a thicker boundary layer and thus a reduced transfer velocity (McKenna and McGillis, 2004).

Surfactants are enriched in the sea surface microlayer in the worlds oceans (Wurl et al., 2011) over a wide range of wind speeds as high as $u_{10} = 13\,\mathrm{m\,s^{-1}}$ (Sabbaghzadeh et al., 2017). In the Baltic sea, high surface activities were measured (Schmidt and Schneider, 2011), with a seasonal dependency at a near-shore location. The reduction of the gas transfer velocity due to surfactants has been observed in studies, where the gas transfer velocity was measured in laboratory setups in fresh water with added artificial surfactants (Mesarchaki et al., 2015; Krall, 2013; Lee and Saylor, 2010; Frew et al., 1995), in water sampled from the ocean (Pereira et al., 2018; Schmidt and Schneider, 2011; Frew et al., 1990; Goldman et al., 1988), during field studies (Frew et al., 2004), as well as during field studies where artificial surfactants were released on the ocean surface (Salter et al., 2011; Brockmann et al., 1982). Gas transfer is found to be highly variable, with a reduction of up to 60 % under surfactant influence.

The gas transfer velocity $k$ of sparingly soluble gases is commonly parameterized with the friction velocity $u_*$, a measure for momentum input,

$$k = \frac{1}{\beta} u_* Sc^{-\mathrm{n}} \tag{3}$$

with the momentum transfer resistance parameter $\beta$ and the Schmidt number exponent $n$ (Deacon, 1977; Jähne et al., 1979; Coantic, 1986; Jähne et al., 1989; Csanady, 1990). Both the momentum transfer resistance $\beta$ and the Schmidt number exponent $n$ depend on the hydrodynamic properties of the water surface. For a hydrodynamically smooth water surface, e.g. at very low wind speeds or under surfactant influence, the Schmidt number exponent is found to be $n = 2/3$, while for a wavy water surface, $n = 1/2$. For increasing friction velocity, this change from $n = 2/3$ to $n = 1/2$ is found to be smooth, rather than sudden (Jähne et al., 1987; Richter and Jähne, 2011). In addition, this change in the Schmidt number exponent depends also on the contamination of the water surface with surface active material, with the change starting at higher friction velocities and being steeper for a surfactant covered water surface (Krall, 2013).

## 2.2 Fetch and wave age

Another factor influencing the gas transfer velocity, which is disregarded in the widely used wind speed only parameterizations, is the dependency on fetch or the age of the wave field. Earliest indications that the fetch is an important parameter were seen by Broecker et al. (1978), who used an 18 m long wind-wave tank and found almost a doubling of the gas transfer velocity compared to the earlier work by Liss (1973), who used a tank of only 4.5 m length. Wanninkhof (1992) pointed out, that the differences observed between gas transfer measurements in lakes and the ocean might be caused by growing wave fields and thus increasing near surface turbulence over distances as high as a few hundreds of kilometers offshore. Zhao et al. (2003) and Woolf (2005) developed parameterization for the transfer velocity based the breaking-wave parameter (Toba and Koga, 1986) and the whitecap coverage, both of which depend on the fetch. The considerations above indicate that there should be a dependency of the gas transfer velocity on the fetch. But unfortunately there is no solid knowledge because more detailed measurements and theories are lacking.

## 3 Measuring technique

### 3.1 Active thermography

The active controlled flux technique (ACFT) can be used to measure gas transfer velocities under laboratory as well as under field conditions with a high temporal (minutes) and spatial (meters) resolution, using heat as a proxy tracer. A carbon dioxide laser with an scanning optic is used to deposit energy directly to the water surface. An infrared camera measures the resulting heating. For this study the system theory approach proposed in Jähne et al. (1989) was used. In this approach, the laser is switched on and off with changing frequencies. At low laser forcing frequencies the water surface will reach the thermal equilibrium, resulting in a constant heating. At higher forcing frequencies this equilibrium is not reached and the measured amplitude is damped. Using Fourier analysis to determine this amplitude damping in dependency of the laser forcing frequency, the time to reach the thermal equilibrium, which corresponds to the response time of the system, is calculated. It is linked to the transfer velocity by

$$k_{\mathrm{heat}} = \sqrt{\frac{D_{\mathrm{heat}}}{\tau}} \quad \text{or} \quad \tau = \frac{D_{\mathrm{heat}}}{k_{\mathrm{heat}}^2}, \tag{4}$$

(Jähne et al., 1987). This analysis technique is particularly suitable for field measurements as it requires no absolute calibration. A more detailed description of the analysis method, the necessary correction for the penetration depth of the infrared camera and the error estimation can be found in Nagel (2014).

### 3.2 Scaling heat transfer velocities to gas transfer velocities

To compare the measured transfer velocities of heat to the transfer velocities of a gas like $CO_2$, Schmidt number scaling is applied,

$$k_{\mathrm{gas}} = k_{\mathrm{heat}} \left( \frac{\mathrm{Sc}}{\mathrm{Pr}} \right)^{-n}, \tag{5}$$

where $k_{\mathrm{gas}}$ and $k_{\mathrm{heat}}$ are the transfer velocities for the gas and heat, respectively. The Schmidt number $\mathrm{Sc} = \nu/D_{\mathrm{gas}}$ and the Prandtl number $\mathrm{Pr} = \nu/D_{\mathrm{heat}}$ are given by the kinematic viscosity of the water divided by the diffusion coefficient of the gas and of heat in water, respectively. The Schmidt number exponent $n$ varies between $n = 2/3$ for a flat and $n = 1/2$ for a wavy water surface (Jähne et al., 1987; Richter and Jähne, 2011; Krall, 2013). Schmidt number scaling is used to provide a value for the gas transfer velocity, which is independent of the specific measurement technique or tracer.

However, using heat as a proxy for a gas tracer has one significant drawback. Diffusion of heat is about one hundred times faster than diffusion of a dissolved gas in water. Because of this, any uncertainty in the Schmidt number exponent $n$ leads to a relative large uncertainty for the heat transfer velocity scaled to a gas transfer velocity. It is generally given by

$$\frac{\Delta k}{k_{\mathrm{gas}}} = \ln \left( \frac{\mathrm{Sc}}{\mathrm{Pr}} \right) \Delta n. \tag{6}$$

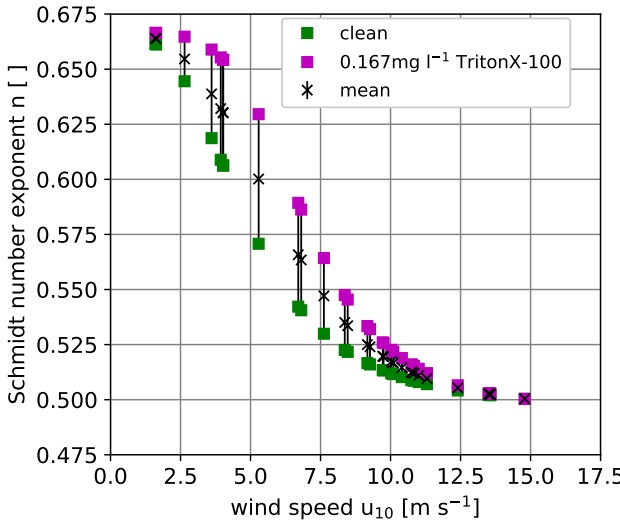

**Figure 1.** Possible ranges of Schmidt number exponents for a clean and surfactant covered water surface as a function of the wind speed as inferred from experiments in the Heidelberg *Aeolotron* wind-wave tank (Krall, 2013) for the wind speeds encountered during this study. Friction velocities measured in the *Aeolotron* were taken from Bopp (2011) and converted to the wind speed in $10\,\mathrm{m}$ height using the drag coefficient parameterization by Edson et al. (2013). To scale the heat transfer velocities measured in the present work, the mean values of the Schmidt number exponent were used.

where $\Delta k$ and $\Delta n$ are the absolute uncertainties for the transfer velocity and the Schmidt number exponent, respectively. For the whole expected range of $n = 2/3$ to $1/2$, $\Delta_n = \pm 0.083$ (Fig. 1) and $\mathrm{Sc}/\mathrm{Pr} \approx 600/9$, the relative scaling error is $\pm 35\,\%$. This is quite a large uncertainty.

In the past decade, several studies found deviations between the Schmidt number scaled heat and the simultaneously mea-
sured gas transfer velocities (Asher et al., 2004; Atmane et al., 2004; Zappa et al., 2004; Jessup et al., 2009). However, a more recent study by Nagel et al. (2015) showed that using a model independent analysis method, as proposed by Jähne et al. (1989) and the correct Schmidt number exponent results in a good agreement.

For field measurements, the importance of using a Schmidt number exponent, depending on the water surface condition is also highlighted in Esters et al. (2017), who relate the gas transfer velocity to the turbulent energy dissipation rate.

Currently, there are no measurement techniques available to measure the Schmidt number exponent in the field with the same temporal resolution as the heat transfer measurements. Therefore, the scaling in the present work was done using Schmidt number exponents measured in the Heidelberg *Aeolotron* wind wave tank (Krall, 2013), as opposed to Schimpf et al. (2011), who used a fixed Schmidt number exponent of $1/2$. In Krall (2013), Schmidt number exponents were measured with different concentrations of the surface active material (surfactant) *Triton X-100*. The mean of the Schmidt number exponent of the two
extreme cases presented in Krall (2013), corresponding to clean water and water with $167\,\mu\mathrm{gl}^{-1}$ Triton X-100, respectively,

was used to scale the heat transfer velocities to gas transfer velocities (see Fig. 1) to account for possible contamination of the water surface with surface active material. The difference between the mean and both extreme values of the Schmidt number exponent was used as the uncertainty of the Schmidt number exponent. Since the *Aeolotron* wind-wave tank is an annular facility, it has virtually unlimited fetch, comparable with open ocean conditions. Due to the lack of simultaneously measured Schmidt number exponents in the field, this approach is more realistic than using $n = 1/2$ for all encountered wind conditions disregarding a potentially smooth condition ($n = 2/3$) of the water surface. The approach used here reduces the uncertainty of $\Delta n$ from $\pm 0.083$ to $< \pm 0.030$ (Fig. 1). The resulting relative uncertainty of $k$ is then $\Delta k / k < \pm 13\%$. Another source of uncertainty lies in transferring the lab measurements of the Schmidt number exponent to the field conditions, since in the lab, the friction velocity $u_*$ is measured (Bopp, 2011) as opposed to the wind speed in 10 m height which is commonly measured in the field. To convert lab measurements to field conditions, the drag coefficient, $C_D = u_*^2 u_{10}^{-2}$ taken from Edson et al. (2013) was used.

## 4  Measurements

### 4.1  Baltic Sea campaigns 2009 and 2010

Three ship campaigns were conducted in 2009 and 2010. Figure 2 show the tracks of these three cruises. The first one (Alkor Cruise 336, Schmidt (2009)) took place from 25 April 2009 until 7 May 2009 on the German research vessel FS *Alkor*. It included measurements north-west of Rügen and the Gotland Sea. The second cruise on the same vessel (Alkor Cruise 356, Schneider (2010)), between 30 June and 13 July 2010 included measurement stations spread over the whole Baltic Sea. The third cruise took place on the Finnish research vessel RV *Aranda* from 14 September until 19 September 2010. Due to the stormy weather conditions, most measurements were conducted in the Finnish archipelago and only two measurements were conducted under open ocean conditions in the Gulf of Finland.

### 4.2  Experimental setup on ship

To use the CFT method described in Sec. 3.1, a $CO_2$-Laser (Firestar f200, Synrad, Inc.) was used to heat the water surface. A scanning system (Micro Max 671, Cambridge Technology, Inc.) was used to widen the laser to create a heated patch on the water surface. The temperature response of the water surface was recorded with an infrared camera (CMT 256, Thermosensorik). Laser, scanner and camera are synchronised by custom electronics. A water tight box, including the IR laser, the IR camera and the electronics was installed on rails on top of an aluminum cradle at the bow of the research vessels. During transit times the box was retracted and fixed over the vessel, while it was moved over the ocean during measurement times. A more detailed description of all used instruments is given in Nagel (2014).

Measurements were only conducted at stations were the vessel was standing at one position. Nevertheless due to currents the water surface moved relative to the ship. As direct sun irradiation disturbs the infrared signals, most measurement were conducted during night time or on cloudy days. Nevertheless, reflections of thermal signature of the sky and the ship itself

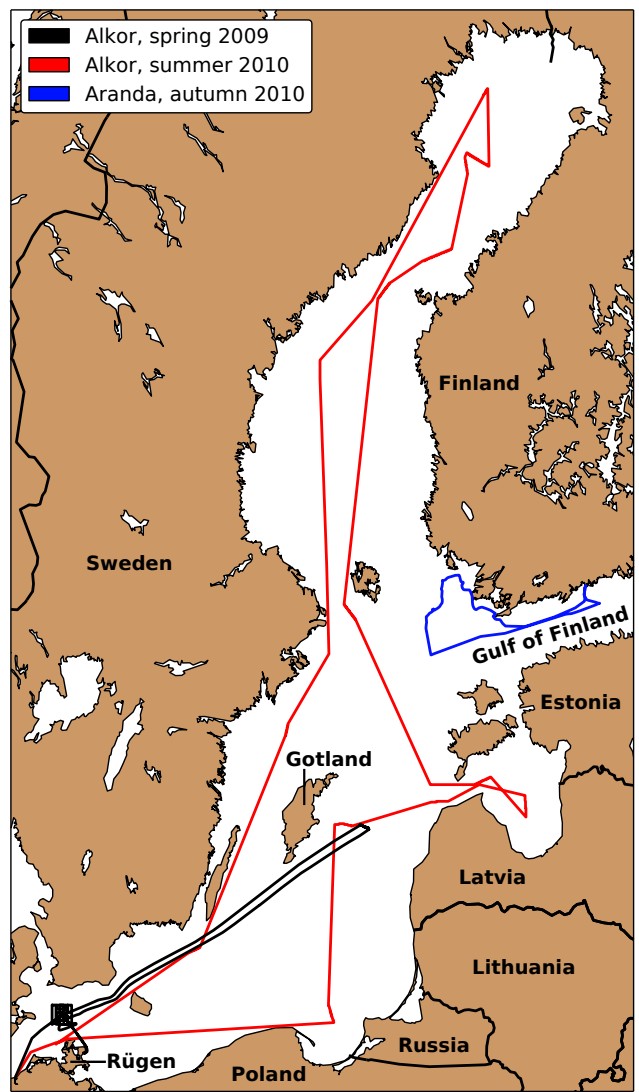

**Figure 2.** Map of the Baltic Sea. The tracks of the three cruises are shown.

can not be avoided. However, the periodic forcing of the heat flux as described in Sect. 3.1, suppresses these effects (lock-in technique).

Wind speed measured in 10 m height was provided by the weather station of each vessel. On FS *Alkor*, one minute mean wind speeds were stored only for the times during which measurements with the ACFT were performed. On RV *Aranda*, ten
5 second mean values were stored for the whole duration of the cruise. During data processing, averages of the stored values were calculated for the times during which the respective ACFT measurements were performed.

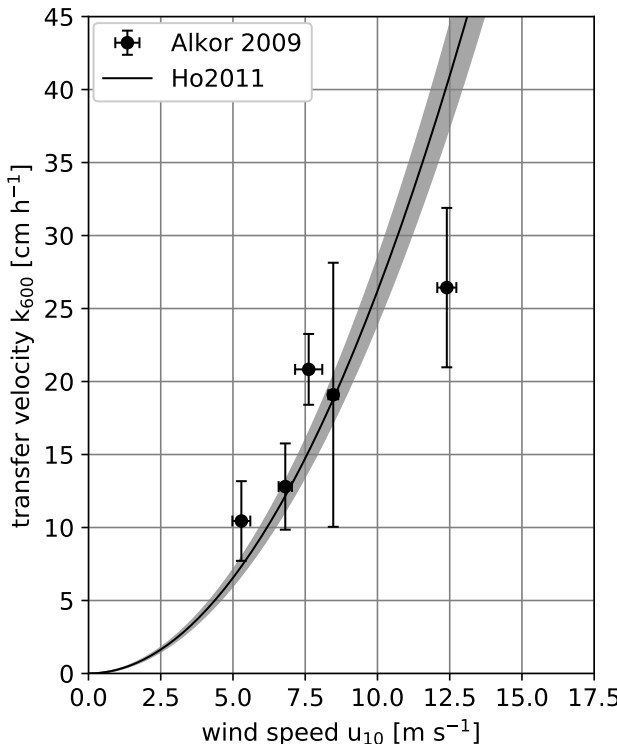

**Figure 3.** Measured $k_{600}$ transfer velocities plotted against the wind speed of the FS *Alkor* Spring 2009 cruise. For comparison the best fit of Ho et al. (2011), Eq. 2 is added.

## 5 Results

### 5.1 Measured transfer velocities

First results of the cruise in 2009 are already published in Schimpf et al. (2011). For this study a re-evaluation with slight differences in the correction of the penetration depth of the infrared camera was done. Also, the improved Schmidt number scaling described in Sect. 3.2 was used, while Schimpf et al. (2011) used n=1/2 for all conditions. The obtained heat transfer velocities are given in Tab. A1. Figure 3 shows the measured transfer velocities, scaled to a Schmidt number of 600. To compare the results with other field measurements the parameterization by Ho et al. (2011), which parameterizes the transfer velocity with the wind speed is also shown. This parameterization was chosen for comparison, since it is one of the few in which a margin of uncertainty is included (gray band in Fig. 3).

Figure 4 shows the measured heat transfer velocities against the wind speed for the *Alkor* campaign in 2010 in comparison with the parameterization by Ho et al. (2011). Schmidt number scaling was done with the same method as for the *Alkor* 2009 data set. During most of the FS *Alkor* campaign in 2010 the wind speeds were rather low. At low wind speeds, the response

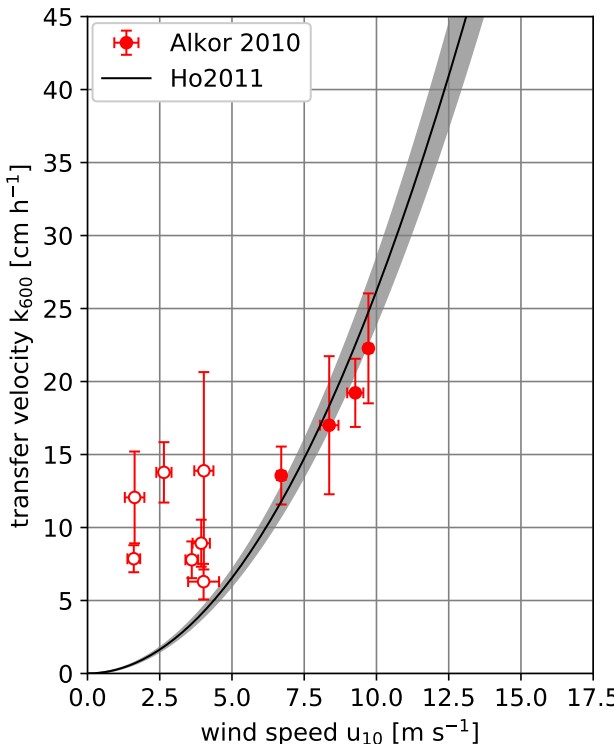

**Figure 4.** Measured $k_{600}$ transfer velocities plotted against the wind speed of the FS *Alkor* Summer 2010 cruise. Conditions for which the measured transfer velocity is likely overestimated are marked with open circles and will not be used for further analysis. For comparison the wind speed parameterization taken from Ho et al. (2011) is added.

time of the water surface is very long, as it increases with the square of the inverse transfer velocity (Eq. 4). The time a water parcel stays in the heated patch (residence time) is limited due to surface currents and the movement of the ship relative to the water surface. In the thermal equilibrium, the heat energy deposited on the water surface by the laser equals the energy removed from the surface by processes driving heat transfer, which results in a constant water surface temperature. Only if the residence

5   time is longer than the response time, the water surface reaches the thermal equilibrium. Otherwise a lower temperature and therefore a higher amplitude damping will be observed, which leads to an overestimation of the measured transfer velocities. The residence times were estimated from the infrared images themselves by measuring the time a single structure stayed in the heated patch. All measurements with wind speeds of $4\,\mathrm{ms}^{-1}$ and below are not reliable, because the estimated residence times were found to be too long. Therefore they will be excluded from further analysis.

10      This highlights the difficulties of measuring gas transfer velocities at very low wind speeds. However, difficulties also exist with other approaches to measure the gas transfer velocity in the field, such as dual tracer studies, where the time scales required for measurements are very long at low wind speeds, and sufficiently long periods of low winds are rarely encountered.

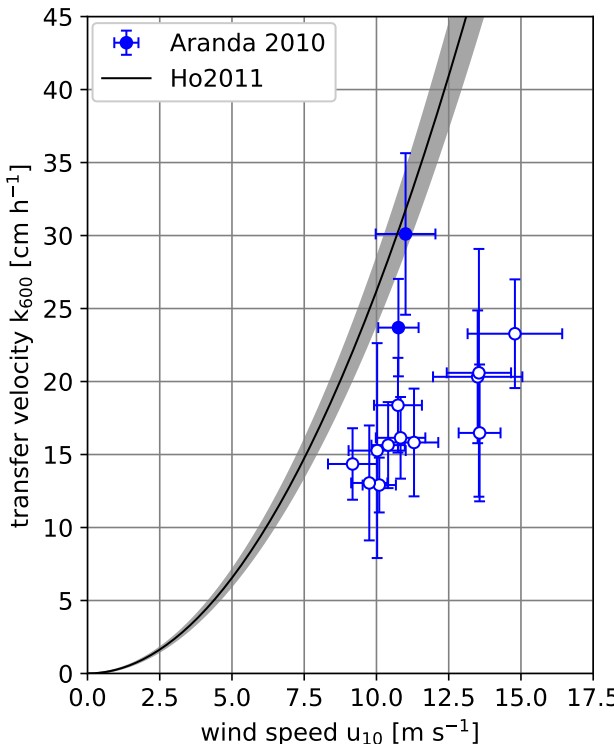

**Figure 5.** Measured $k_{600}$ transfer velocities plotted against the wind speed of the RV *Aranda* Fall 2010 cruise. The filled circles show the open ocean measurements, while the open circles are data from the archipelago. For comparison, the wind speed parameterization by Ho et al. (2011) is also shown.

The heat transfer velocities scaled to Sc=600 measured on RV *Aranda* in 2010 are shown in Fig. 5. The transfer velocities measured in the shielded archipelago are significantly lower than the ones measured under open ocean conditions.

## 5.2 Comparison with other field and laboratory data

Figure 6 shows a comparison between the measured transfer velocities and the empirical parameterization of Ho et al. (2011). The measurements from the *Alkor* 2009 and *Alkor* 2010 cruises coincide within the error margins with the empirical parameterization by Ho, except for the value at the highest wind speed, which is approx. 40% lower. The two open ocean measurements during the RV *Aranda* cruise 2010 are slightly lower than the empirical parameterization, but still close to it.

This is, however, not the case for the RV *Aranda* cruise measurements in the shielded archipelago. The measured values are significantly lower. On average, the values are only about one half of the transfer velocities predicted by the empirical parameterization. There are three possible explanations for this finding: bubble-mediated transfer, fetch or wave-age and surfactants. In the following sections, these possible explanations will be discussed in detail.

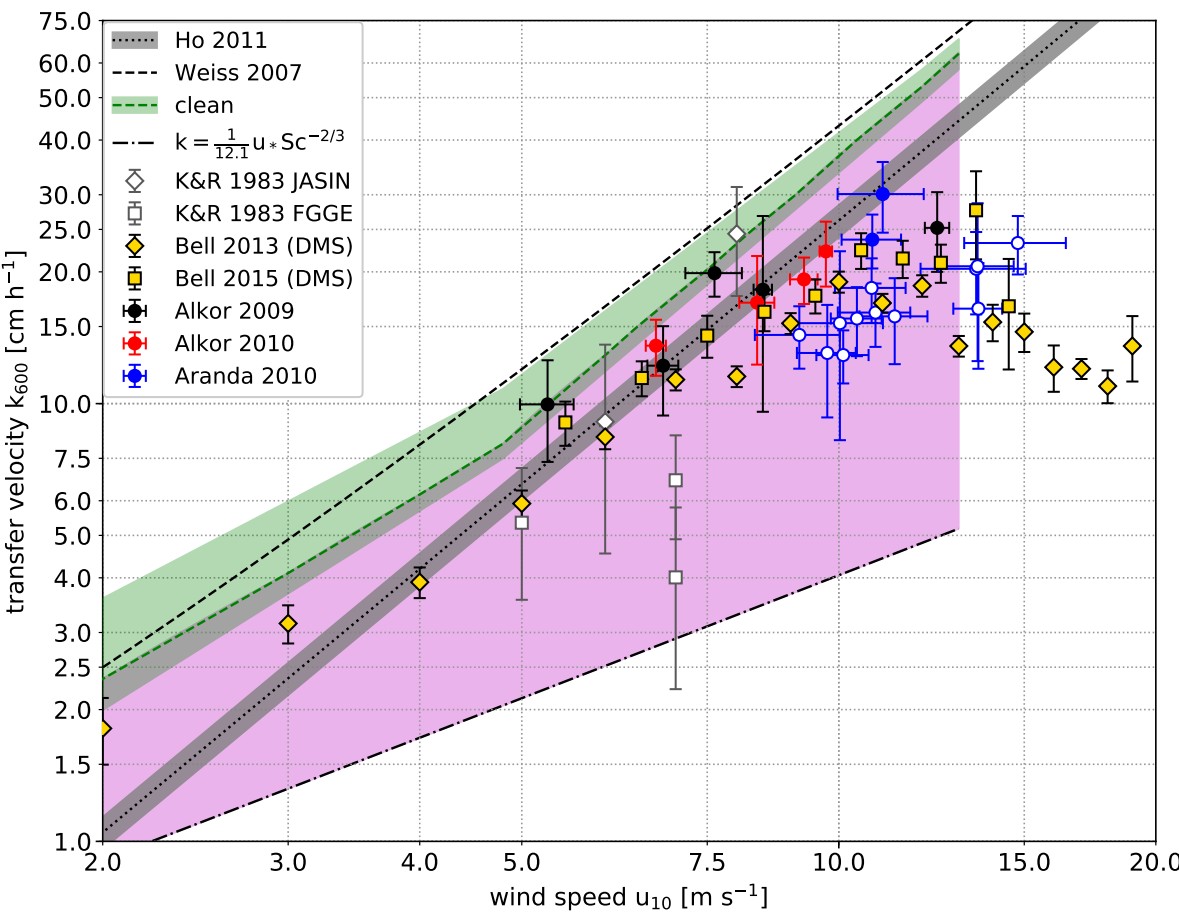

**Figure 6.** Comparison of scaled heat transfer velocities measured in the Baltic Sea and gas transfer velocities measured in the Heidelberg *Aeolotron* wind-wave facility with a clean water surface (green shaded area). The measurements on RV *Aranda* in 2010 which were made under open ocean conditions (i.e. with a virtually unlimited fetch) are marked with filled circles, while the fetch limited measurements in the archipelago are marked with open circles. Also shown is the lower limit for a smooth water surface, Eq. 3. The region between the transfer velocities measured with a clean water surface as the upper boundary and the values for a smooth water surface as the lower boundary for possible transfer velocities is shaded in magenta. Also shown are the data set from the North Atlantic of Kromer and Roether (1983) (K&R1983) using the radon deficit method, DMS eddy covariance measurements (Bell et al., 2013, 2015) and the parameterization of previous Baltic Sea gas transfer measurements by Weiss et al. (2007). The individual data points in Weiss et al. (2007) and Rutgersson et al. (2008) scatter too strongly to be shown here. Also shown is the parameterization by Ho et al. (2011).

### 5.2.1 Bubble-mediated transfer

It is known that active thermography misses the contribution by bubbles to the transfer, see Sect. 2. Because of the high solubility of dimethylsulfide (DMS), this tracer's gas transfer has almost no bubble-induced component and the transfer velocities

of DMS measured by Bell et al. (2013, 2015) have values very similar to ours, indeed (Fig. 6). Another observation, which supports this argument are the higher $CO_2$ gas exchange values ($CO_2$ has a significantly lower solubility than DMS with a higher expected bubble-induced contribution) measured in the Baltic Sea by Weiss et al. (2007) and Rutgersson et al. (2008). We only show the combined linear/quadratic parameterization by Weiss et al. (2007), since Rutgersson et al. (2008) does not give a parameterization.

A very helpful hint comes, however, from laboratory experiments, which suggest that this explanation is not correct. No evidence for a significant bubble contribution to gas transfer was found in a laboratory study (Krall, 2013) up to the highest wind speed used in that study ($\approx 12\,\mathrm{m\,s^{-1}}$), although tracers with solubilities much lower than $CO_2$ (dimensionless solubility $\alpha \approx 0.7$) and DMS ($\alpha \approx 11.2$) were used, including $N_2O$ ($\alpha \approx 0.5$), trifluoromethane ($\alpha \approx 0.26$), and pentafluoroethane ($\alpha \approx 0.07$). In another study, Nagel et al. (2015) found no differences between gas transfer velocities of $N_2O$ and heat transfer velocities for wind speeds as high as $12\,\mathrm{m\,s^{-1}}$, which indicates that bubble contribution for both, the transfer heat and that of $N_2O$ is not significant.

### 5.2.2 Fetch and wave-age effects

A second explanation would be the effect of fetch or, equivalently, quickly varying wind conditions with young wave ages. This effect has almost not be studied so far. Only recently, Kunz and Jähne (2018) showed with active thermography measurements in the Heidelberg Aeolotron that, at very short fetches and low wind speed, the gas transfer velocity is significantly higher than at infinite fetch. This finding is supported by an old data set, which constitutes the most diligently measured gas transfer velocities using the Radon deficit method (Kromer and Roether, 1983; Roether and Kromer, 1984). One part of this data set was measured during the JASIN cruise in the North Atlantic with highly varying wind speeds. The measured gas transfer velocities are higher or as high as predicted by the empirical parameterization. However, the transfer velocities measured during the FGGE cruise with constantly blowing trade winds are significantly lower. One value is three times lower than predicted by the empirical parameterization of Ho. These measurements clearly indicate that even at the open ocean (i. e. without fetch limitations) there will be significant differences in the gas transfer velocity. The data suggests that this effect may be as large as a factor of five.

Surprisingly, the thermographic measurements in the Baltic Sea show just the opposite dependency. In the shielded archipelago with probably short fetches, the transfer velocities are lower and not higher. Thus fetch dependency does not seem to be the correct explanation in this case at rather high wind speeds, where also the Aeolotron data by Kunz and Jähne (2018) show no significant fetch dependency.

### 5.2.3 Surfactants

The third and most likely reason for the lower gas exchange rates during part of the Aranda 2010 cruise is a reduction of the transfer velocity by surface films. The reduction of about a factor of two is consistent with earlier measurements discussed in Sect. 2.1.

At this point it is helpful to compare the field data with laboratory data augmented by physical arguments about the mechanisms of air-sea gas transfer. However, a direct comparison is physically not valid, because the conditions concerning the wave field and surface contamination will be different. Despite that, laboratory data is suited to explore the possible upper and lower limits of the gas transfer velocity at a given wind speed. The Heidelberg *Aeolotron* laboratory has a virtually unlimited fetch

due to its annular shape, so it may resemble the ocean conditions in the best possible way. The gas transfer velocities measured when the water surface in the Aeolotron was carefully cleaned by skimming the top layer of the water before the start of each measurement to remove surface active material, can be considered to be the upper limit (green shaded area in Fig. 6). Those gas transfer velocities were measured with the method described in Mesarchaki et al. (2015) and are published in Krall (2013). In the green shaded area, the increase in the gas transfer velocities at low wind speeds and short fetches observed by Kunz and

Jähne (2018) is taken into account, too.

The lower limit for possible gas transfer velocities is given by the prediction of Deacon (1977) (Eq. 3 with n=2/3 and $\beta =12.1$) for a smooth water surface. These values have been confirmed by measurements in a small annular wind/wave facility, when the water surface was covered by surfactants (Jähne et al., 1979). The highest friction velocity in water at which the water surface remained smooth and without wind waves in this facility was $1.4\,\mathrm{cm\,s^{-1}}$ corresponding to a smooth water

surface up to a wind speed of $u_{10} \approx 13\,\mathrm{m\,s^{-1}}$. This is supported by the findings of Sabbaghzadeh et al. (2017), who measured surfactant enrichment in the sea surface microlayer up to $u_{10} \approx 13\,\mathrm{m\,s^{-1}}$ as well.

The region between these upper and lower bounds for gas transfer is shaded in a magenta color in Fig. 6. This difference between highest and lowest possible gas transfer velocities alone indicates that the gas transfer is highly variable and not only dependent on wind speed alone.

All shown field data based on mass balance methods, eddy covariance and active thermography are compatible with this shaded region of possible gas transfer velocities. The parameterization of $CO_2$ measured with the eddy covariance technique in the Baltic Sea according to Weiss et al. (2007) is slightly higher than the upper limit resulting from laboratory measurements. Because of the high scatter of these data, some individual measurements are even much higher. This means that we still see discrepancies between measurements based on mass balance (now including also active thermography) and eddy covariance

measurements, although they are not as bad as in the early days of eddy covariance measurements (Broecker et al., 1986).

## 6   Conclusions and outlook

Heat exchange measurements were conducted in the Baltic Sea during three different campaigns using the active controlled flux technique. The measured heat transfer velocities, scaled to gas transfer velocities using realistic Schmidt number exponents, show high variability even at the same wind speed. New is that even at high wind speeds in the range of 8 to $15\,\mathrm{m\,s^{-1}}$

significantly lower gas transfer velocities were measured, which were about a factor of two lower than the average transfer velocities measured by the dual tracer technique and parameterized by the relation of Ho et al. (2011). Based on arguments from several lab studies, the influence of surfactants is the most likely reason for variability of the gas transfer velocity under

the environmental conditions for the thermographic measurements in the Baltic sea. But a possible influence of fetch and bubbles on these measurements cannot completely be ruled out.

Therefore this study clearly indicates that a better understanding of air-sea gas transfer urgently requires more systematic measurements of the effects of bubbles, fetch (or the age of the wave field) and surfactants. In the field the most promising approach is eddy covariance measurements together with active thermography. For laboratory measurements some serious limitations must be overcome. One is the fetch gap. In linear facilities only very short fetches can be studied, which are no longer than the maximum length of the water tunnel in the facility. Even at these short fetches, significant variations of the gas transfer rate can be measured. This has recently been demonstrated by Kunz and Jähne (2018) using active thermography.

In order to increase the fetch range available in the lab, gas exchange measurements could be performed in annular facilities under unsteady wind speed conditions. In the Heidelberg *Aeolotron* it is possible to switch on the wind in a few seconds, while it takes several minutes for the wave field to develop to a stationary state. Unfortunately, it is very hard to make gas exchange measurements with a temporal resolution of below a minute using conventional mass balance techniques.

A very promising technique for fast measurements of gas transfer is the recently developed mass boundary layer imaging technique (Kräuter et al., 2014; Kräuter, 2015). Using this technique will enable the measurement of the gas transfer velocity simultaneously and in the same footprint as the heat transfer velocity. This will allow a direct comparison as well as in-depth studies of the physical mechanisms governing air-sea gas and heat transfer.

## Appendix A: Numerical values of the measured transfer velocities

Tables A1, A2 and A3 give the numerical values of the measurements conducted during the cruises in the Baltic Sea.

**Table A1.** Measured heat transfer velocities $k_{heat}$ in dependency of time, position, wind speed and water and air temperature for the measurements on FS *Alkor* in 2009. Furthermore the Prandtl number Pr, the Schmidt number exponent $n$ and the scaled transfer velocity $k_{600}$ are given. The given times are approximate starting times in UTC. Each measurements lasted about $20\,\text{min}$.

| number | date | time | position | | $u_{10}$ | $T_{water}$ | $T_{air}$ | $k_{heat}$ | Pr | n | $k_{600}$ |
|---|---|---|---|---|---|---|---|---|---|---|---|
| | yyyy/mm/dd | hh:mm | N | E | $[\text{m s}^{-1}]$ | $[^oC]$ | $[^oC]$ | $[\text{cm h}^{-1}]$ | | | $[\text{cm h}^{-1}]$ |
| A1 | 2009/04/28 | 19:55 | 55.002 | 13.169 | 8.47±0.17 | 7.3 | 10.8 | 158.6±74.8 | 10.38 | 0.534±0.012 | 18.2±8.6 |
| A2 | 2009/04/30 | 02:30 | 55.122 | 13.103 | 12.4±0.33 | 7.4 | 8.2 | 195.9±40.4 | 10.38 | 0.505±0.001 | 25.2±5.2 |
| A3 | 2009/05/01 | 20:05 | 56.389 | 17.591 | 5.29±0.31 | 5.7 | 6.4 | 109.8±25.6 | 11.0 | 0.6±0.029 | 10.0±2.6 |
| A4 | 2009/05/02 | 20:20 | 57.337 | 20.016 | 6.81±0.23 | 6.2 | 8.0 | 117.3±24.8 | 10.81 | 0.563±0.023 | 12.2±2.8 |
| A5 | 2009/05/03 | 20:45 | 57.366 | 19.904 | 7.62±0.47 | 6.5 | 7.9 | 179.8±16.8 | 10.7 | 0.547±0.017 | 19.9±2.3 |

**Table A2.** Measured heat transfer velocities $k_{\text{heat}}$ in dependency of time, position, wind speed and water and air temperature for the measurements on FS *Alkor* in 2010. Furthermore the Prandtl number Pr, the Schmidt number exponent $n$ and the scaled transfer velocity $k_{600}$ are given. The given times are approximate starting times in UTC. Each measurements lasted about $20\,\text{min}$. Measurements marked with an asterisk (*) are considered unreliable, see Sect. 5.1.

| number | date | time | position | | $u_{10}$ | $T_{\text{water}}$ | $T_{\text{air}}$ | $k_{\text{heat}}$ | Pr | n | $k_{600}$ |
|--------|------|------|----------|----------|----------|----------|----------|----------|------|---|----------|
| | yyyy/mm/dd | hh:mm | N | E | $[\text{m}\,\text{s}^{-1}]$ | $[^oC]$ | $[^oC]$ | $[\text{cm}\,\text{h}^{-1}]$ | | | $[\text{cm}\,\text{h}^{-1}]$ |
| B1* | 2010/07/02 | 00:05 | 54.951 | 19.233 | 4.0±0.3 | 17.0 | 15.8 | 217.4±103.3 | 7.63 | 0.63±0.024 | 13.9±6.8 |
| B2* | 2010/07/02 | 00:35 | 55.064 | 19.175 | 3.9±0.3 | 16.6 | 15.8 | 139.6±20.9 | 7.74 | 0.632±0.023 | 8.9±1.6 |
| B3* | 2010/07/03 | 06:05 | 57.383 | 19.490 | 1.6±0.2 | 17.9 | 17.7 | 146.9±17.2 | 7.3 | 0.664±0.003 | 7.9±0.9 |
| B4* | 2010/07/03 | 23:05 | 57.658 | 21.653 | 3.6±0.2 | 18.4 | 18.5 | 130.2±17.6 | 7.3 | 0.639±0.02 | 7.8±1.3 |
| B5* | 2010/07/04 | 22:05 | 57.903 | 22.594 | 4.0±0.5 | 19.5 | 20.1 | 103.2±16.7 | 7.09 | 0.63±0.024 | 6.3±1.2 |
| B6 | 2010/07/05 | 20:30 | 59.857 | 19.643 | 6.7±0.1 | 15.2 | 16.4 | 154.9±16.3 | 8.1 | 0.566±0.024 | 13.6±2.0 |
| B7 | 2010/07/08 | 18:50 | 65.215 | 22.638 | 8.4±0.3 | 14.5 | 16.2 | 168.7±46.1 | 8.23 | 0.535±0.012 | 17.0±4.7 |
| B8* | 2010/07/10 | 22:35 | 58.561 | 18.244 | 2.6±0.3 | 18.9 | 20.9 | 249.4±35.7 | 7.19 | 0.655±0.01 | 13.8±2.1 |
| B9* | 2010/07/10 | 23:05 | 58.567 | 18.246 | 1.6±0.3 | 18.9 | 20.4 | 227.3±59.2 | 7.19 | 0.664±0.003 | 12.1±3.1 |
| B10 | 2010/07/11 | 19:15 | 58.567 | 16.240 | 9.7±0.1 | 19.6 | 22.5 | 225.3±37.6 | 6.99 | 0.52±0.006 | 22.3±3.8 |
| B11 | 2010/07/11 | 19:45 | 58.847 | 16.206 | 9.3±0.3 | 19.9 | 22.4 | 198.2±23.0 | 6.99 | 0.524±0.008 | 19.2±2.3 |

**Table A3.** Measured heat transfer velocities $k_{\text{heat}}$ in dependency of time, position, wind speed and water and air temperature for the measurements on RV *Aranda* in 2010. Furthermore the Prandtl number Pr, the Schmidt number exponent $n$ and the scaled transfer velocity $k_{600}$ are given. The given times are approximate starting times in UTC. Each measurements lasted about 20 min. All measurements were conducted in a fetch-limited position with the exception of the two conditions marked with an asterisk (*).

| number | date | time | position | | $u_{10}$ | $T_{\text{water}}$ | $T_{\text{air}}$ | $k_{\text{heat}}$ | Pr | n | $k_{600}$ |
| | yyyy/mm/dd | hh:mm | N | E | $[\text{m s}^{-1}]$ | $[^oC]$ | $[^oC]$ | $[\text{cm h}^{-1}]$ | | | $[\text{cm h}^{-1}]$ |
| --- | --- | --- | --- | --- | --- | --- | --- | --- | --- | --- | --- |
| C1 | 2010/09/15 | 18:05 | 59.899 | 21.502 | 10.4±0.6 | 14.9 | 13.3 | 143.6±25.7 | 8.07 | 0.515±0.004 | 15.6±2.8 |
| C2 | 2010/09/15 | 21:25 | 59.899 | 21.502 | 9.2±0.8 | 14.8 | 13.8 | 137.6±21.8 | 8.1 | 0.525±0.008 | 14.4±2.3 |
| C3 | 2010/09/16 | 04:15 | 59.899 | 21.502 | 13.6±0.7 | 14.9 | 14.1 | 143.6±38.9 | 8.07 | 0.502±0.0 | 16.5±4.5 |
| C4 | 2010/09/16 | 05:30 | 59.899 | 21.502 | 14.8±1.6 | 14.9 | 14.0 | 201.0±30.7 | 8.07 | 0.5±0.0 | 23.3±3.5 |
| C5 | 2010/09/16 | 16:10 | 59.899 | 21.502 | 13.5±1.5 | 14.9 | 13.9 | 177.2±37.8 | 8.07 | 0.503±0.0 | 20.3±4.3 |
| C6 | 2010/09/16 | 17:15 | 59.899 | 21.502 | 13.5±1.1 | 14.9 | 13.7 | 179.5±70.5 | 8.07 | 0.502±0.0 | 20.6±8.1 |
| C7 | 2010/09/16 | 20:55 | 59.893 | 21.486 | 10.0±1.0 | 14.8 | 13.6 | 141.6±65.0 | 8.1 | 0.517±0.005 | 15.3±7.0 |
| C8 | 2010/09/16 | 21:50 | 59.893 | 21.486 | 10.1±0.6 | 14.7 | 14.0 | 119.2±16.3 | 8.12 | 0.517±0.005 | 12.9±1.8 |
| C9 | 2010/09/17 | 04:15 | 59.893 | 21.486 | 10.7±0.8 | 14.5 | 13.7 | 166.2±27.9 | 8.17 | 0.512±0.004 | 18.4±3.1 |
| C10 | 2010/09/17 | 05:25 | 59.893 | 21.486 | 10.8±0.9 | 14.6 | 13.7 | 145.9±24.0 | 8.14 | 0.512±0.003 | 16.1±2.7 |
| C11 | 2010/09/17 | 16:15 | 59.893 | 21.486 | 11.3±0.8 | 14.6 | 13.4 | 141.5±31.4 | 8.14 | 0.51±0.003 | 15.8±3.5 |
| C12 | 2010/09/17 | 19:15 | 59.893 | 21.486 | 9.8±0.6 | 14.5 | 13.6 | 121.6±34.8 | 8.17 | 0.519±0.006 | 13.1±3.8 |
| C13* | 2010/09/18 | 13:05 | 59.378 | 21.441 | 11.0±1.0 | 14.0 | 12.2 | 268.5±49.2 | 8.29 | 0.511±0.003 | 30.1±5.5 |
| C14* | 2010/09/18 | 13:35 | 59.378 | 21.441 | 10.8±0.7 | 13.2 | 11.3 | 209.9±29.4 | 8.49 | 0.512±0.004 | 23.7±3.3 |

*Author contributions.* L.N. was involved in designing the experiments, performed the experiments and calculated the heat transfer velocities from the IR images. K.E.K. calculated the $k_{600}$ values, prepared all figures and tables and served as communicating author. B.J. was involved in designing the experiments and the analytical methods, and outlined the main conclusions. All authors contributed to writing the manuscript.

*Competing interests.* None.

5  *Acknowledgements.* We would like to thank Prof. Dr. Kimmo Kahma and Dr. Heidi Pettersson, Finish Meteorological Institute for the possibility to participate in the Aranda CO2_WAVE10_CTD10/2010 cruise. We would also like to thank Dr. Robert Schmidt and Dr. Bernd Schneider, Institut für Ostseeforschung, Warnemünde for their chief scientist work during the cruises *Alkor* 336 and *Alkor* 356. We are grateful for the assistance onboard by the captains and crews of RV *Aranda* and FS *Alkor*. We would like to thank Dr. Uwe Schimpf and Dr. Günther Balschbach for help with preparing and conducting the measurements as well as logistical support. Financial support for this

10  work by the German Federal Ministry of Education and Research (BMBF) joint project "Surface Ocean Processes in the Anthropocene" (SOPRAN, FKZ 03F0462F, 03F0611F and 03F0622F) within the international SOLAS project is gratefully acknowledged.

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
