# Peer review of "Measurements of air-sea gas transfer velocities in the Baltic Sea"

_Ocean Science, 2018_

## Referee Comment (RC1) · E. S. Saltzman (Referee) · 28 Oct 2018

This paper presents new and reprocessed older field measurements using the active flux thermography technique. These are very interesting observations - the improved analysis/processing methodology is clever and should be much more robust than earlier approaches. In addition, recognition of variable Sc number dependence based on lab studies provides improved interpretation of the measurements. The data are a welcome addition to the literature on in situ air/sea gas flux studies and should be published. This paper is also very clearly written and nicely organized, which is much appreciated by this reviewer. There are a few scientific comments below which I would like the authors to consider in their interpretation (or at least discussion) of the data.

[Figure]

There seems to be a rather strong bias against eddy covariance in this paper - the only comment about more than a decade of new work in that area is rather dismissive and citation-less. So, a reader new to the field would imagine that eddy covariance is not generating insight into air-sea gas transfer (which I think most would agree is not the case). The fact that eddy covariance data are often binned seems like an odd criticism, especially when the dual tracer method (which requires long averaging) is held up as the "gold standard". The uncertainty in a single DMS eddy covariance measurement under favorable conditions is on the order of 25% and one could easily imagine interesting results from simultaneous eddy covariance and active thermography measurements.

I was surprised that so much emphasis in this paper was placed on the dual tracer method because active thermography captures only interfacial flux. Bubble-related transfer is very important for $CO_2$ so if active thermography agrees with dual tracer at intermediate and high winds, then it would seem that some assumption in the interpretation of these methods is wrong. Active thermography should be more similar to eddy covariance measurements of DMS than to a dual tracer fit meant to mimic $CO_2$. Eddy covariance studies of DMS and $CO_2$ clearly show that $CO_2$ fluxes at intermediate and high winds are enhanced by bubble transfer relative to dms (which is controlled mostly by the interfacial flux; for example, Bell et al., 2013; Blomquist et al., 2017). There is a conundrum here - if the dual tracer method gets kco2 right (which it seems to), it must be bubble-enhanced also. So one would expect active thermography to diverge from the dual tracer results at intermediate and higher wind speeds.

Several studies suggest that interfacial gas transfer appears to be limited at higher winds. This is attributed to wave shielding and other wave-related effects demonstrated in the laboratory by Mueller and Veron (2009) and incorporated into gas transfer models by Fairall et al., 2011 and Donelan and Soloviev, 2016. Such processes could be salient here in relating active thermography to gas transfer. This is not to say that the arguments in the paper about fetch and surfactants etc. are not very well founded. I

think they are. But the overall premise that the dual tracer and active thermography measurements should measure the same thing seems open to debate. I think this should be considered by the authors and perhaps addressed in the manuscript.

---

## Referee Comment (RC2) · Anonymous Referee #2 · 9 Nov 2018

The article presents a comparison of data obtained during 3 measurement campaigns in the Baltic Sea, using the active thermography method, to investigate the variability of the transfer velocities in field conditions, with data from Ho et al., in which the dual trace method was used. In addition, data from two measurement campaigns using the Radon deficit method were used. During the Baltic cruise heat transfer velocity was measured. The authors scaled heat transfer velocities to gas transfer velocities using different exponent of Schmidt number (determined in laboratory conditions) and compared the measured transfer velocity with the existing parameterization on gas transfer velocity. As a factor influencing the gas transfer, the authors used fetch and surface active material. These are very interesting study which are very useful and fills the gap for our understanding of air-sea interaction but a few more information will be

welcome in order to complete the results. I recommend the article for the publication after a few changes and some more work.

Major comments: 1. The purposes of the manuscript is not clearly formulated. The title suggests that the purpose was to demonstrate how the air-sea gas transfer varies in the Baltic Sea, but inside the text suggests other objectives. Is the objective to compare active thermography method with dual trace method? Or is the purpose to prove that fetch and surface active material have a significant effect on the gas transfer velocity. Or is the purpose of proving that in the transfer velocity study should we use the Schmidt exponent dependently on water conditions? 2. The article in its current form-short, specific information on a given topic, without any extensive descriptions of the transfer velocity - is attractive for scientists who are interested in this topic. But for scientists who are not familiar with this topic, this article may be embarrassing, because they will learn specific results from the use of a given measurement technique, but will not know anything about why this is happening. This article is of a purely technical nature rather than a scientific article. It is important to add more information about the different methods that are used for study transfer velocity or more information about the gas transfer velocity itself and in the Baltic Sea. 3. Here are more specific comments: a. The Introduction should be extended about information described above. A few sentences about various technique to study gas transfer. What is the ACTF method characterized. Some more information about correlation k with u*. Perhaps more information about variability of air-sea gas transfer velocity in the Baltic Sea, as the title suggested. b. More information why mainly wind speed is taken into consideration in air-sea study and why we should add more factors. Not only write the other factors. c. P2L6 ...the active controlled flux (CFT)... d. P2L 9 add references after: A wealth of studies (references) have shown.... e. P3L27..the active controlled flux (CFT) f. Please exchange para 4.1 with 4.2 for better organization, as you mention in presence para 4.1 cruises which are introduce in presence para 4.2 g. P7L29 - ...from 25 April 2009 until 7 May 2009 on the German....
h. P9L11 During most of the FS Alkor campaign in 2010. . ... this should be after Fig. 5 where you introduce this cruise

i. The results and conclusion are very short when they are the most important part of the article. Maybe comparison with other data from the Baltic Sea.

j. We know that at higher and lower wind speed gas transfer are limited so maybe more about that.

---

## Referee Comment (RC3) · Anonymous Referee #3 · 19 Dec 2018

The paper presents results from three field campaigns in the Baltic Sea where the ACFT approach were used to study heat transfer velocities. These velocities could subsequently be scaled to gas transfer velocities by using previous work from a laboratory study in the Aelotron facility. Additionally, the Schmidt number exponent where allowed to vary depending on actual environmental conditions which is a more physical approach compared to only using a fixed exponent.

General

The main results highlight that wave age might be an important parameter to take into account when parameterizing the gas transfer velocity. This might perhaps be the reason why commonly used parameterizations for lakes differ from ocean measurements (e.g. Cole and Caraco , 1998; Podgrajsek et al. 2015)? For lakes the wind speed

dependence seems less pronounced, whereas other processes such as water side convection likely is more important compared to oceans.

Additionally, the results also highlight that surface active material might be very important to consider when parameterizing k. This is a very interesting finding and is in line with other recent findings. As such I think this study is highly motivated with relevant results.

Major comments

My only major comment is the heavy focus on comparing with the dual tracer technique. The ACFT approach has a completely different 'time constant' relating k to more or less instantaneous wind with a footprint of a few square meters, whereas the dual tracer technique is quite the opposite. I think a more proper comparison would be with eddy covariance based results.

Minor comments

Page 2. Line 12: Waterside convection is also a process which might influence the transfer velocity

Page 2, paragraph starting at line 25, the recent paper by Pereira et al. 2018, would also be good to include here.

Page 5, figure 1: how sensitive are your calculated transfer velocities to the variation Sc?

Page 6, lines 25-26: how long averaging period did you use?

Page 6, line 27, I think section 4.2 suits better before the current 4.1 section.

Page 8, figure 3: I would suggest comparing with an EC based parameterization instead. Additionally, Ho et al. use a 10-min mean wind speed, what averaging period are you using for your wind speeds?

Page 8-9, line 1: please specify what you mean by "the response time of the system is very high", what is meant by "high" here, how long time is this?

Page 9, line 3: similar comment, please specify what the typical response time of the water surface you refer to

Page 9, line 5: again, how long are the residence times estimated from the IR images.

Page 12, line 10: The Aelotron has already been introduced in the text, no need for a second introduction here.

References

Cole, J. J., and N. F. Caraco (1998), Atmospheric exchange of carbon dioxide in a low‐wind oligotrophic lake measured by the addition of SF6, Limnol. Oceanogr., 43(4), 647–656.

Podgrajsek, E., E. Sahlée, and A. Rutgersson (2015), Diel cycle of lake-air CO2 flux from a shallow lake and the impact of waterside convection on the transfer velocity, J. Geophys. Res. Biogeosci.,120,29–38

Pereira, R., I. Ashton, B. Sabbaghzadeh, J .D. Shutler, R. C. Upstill-Goddard (2018), Reduced air-sea CO2 exchange in the Atlantic Ocean due to biological surfactants, Nature Geoscience, 11, 492-496

---

## Author Comment (AC1) · 22 Jan 2019

**Authors' response to 'Reviewer comment for os-2018-108' by E.S. Saltzman**

We would like to express our gratitude to E.S. Saltzman for his thorough and helpful review. Our answers are given in the table below.

| Reviewer's comment | Authors' response |
|---|---|
| There seems to be a rather strong bias against eddy covariance in this paper - the only comment about more than a decade of new work in that area is rather dismissive and citation-less. So, a reader new to the field would imagine that eddy covariance is not generating insight into air-sea gas transfer (which I think most would agree is not the case). The fact that eddy covariance data are often binned seems like an odd criticism, especially when the dual tracer method (which requires long averaging) is held up as the "gold standard". The uncertainty in a single DMS eddy covariance measurement under favorable conditions is on the order of 25% and one could easily imagine interesting results from simultaneous eddy covariance and active thermography measurements. | It was in no way our intention to proclaim the dual tracer technique to be the 'gold standard'. To put our measured gas transfer velocities into perspective, we chose to compare them with the Ho. et al. 2011 parameterization. We decided against using more than this one parameterization, since the many available parameterizations for the transfer of $CO_2$ for both the dual tracer technique and the eddy covariance technique are very similar in the range of wind speeds we studied. The Ho. et al. 2011 parameterization was chosen because it is one of a few parameterizations where a confidence interval is given. Our reasoning would not change if we used a parameterization based on eddy covariance of $CO_2$ to compare our data with. We have changed the manuscript to include an eddy covariance based CO2-transfer parametrization to Fig. 6 and also included some eddy covariance DMS measurements. We have extended the discussion in the results section to also discuss the mentioned additions to Fig. 6. |
| | We mentioned that binning is commonly done for eddy covariance data sets to put the temporal resolution of the ACFT into perspective, not to criticize eddy covariance. |
| | We agree that simultaneous measurements of eddy covariance and ACFT would be very valuable. |
| I was surprised that so much emphasis in this paper was placed on the dual tracer method because active thermography captures only interfacial flux. Bubble-related transfer is very important for CO2 so if active thermography agrees with dual tracer at intermediate and high winds, then it would seem that some assumption in the interpretation of these methods is wrong. Active thermography should be more similar to eddy covariance measurements of DMS than to a dual tracer fit meant to mimic CO2. Eddy covariance studies of DMS and CO2 clearly show that CO2 fluxes at intermediate and high winds are enhanced by bubble transfer relative to dms (which is controlled mostly by the interfacial flux; for example, Bell et al., 2013; Blomquist et al., 2017). There is a conundrum here - if the dual tracer method gets kco2 right (which it seems to), it must be bubble-enhanced also. So one would expect active thermography to diverge from the dual tracer results at intermediate and higher wind speeds. | Iwano et al. 2013 (CO2, solubility 0.8) and Krall&Jähne2014 (two tracers with solubility 1 and 3.2) found the measured gas transfer velocities to be compatible with the theoretical prediction for pure interfacial transfer of $$k = \beta^{-1} u_* Sc^{-n}$$ up to wind speeds of around 30-35 m/s. Both of these studies were done in a wind-wave tank using fresh water, in which the bubble size distribution differs from sea water. However, gas transfer velocities measured in a hurricane (McNeil&D'Asaro2006, O2 with solubility of 0.03) agree well with the transfer velocities measured in both lab studies mentioned above. From this line of evidence we can infer that bubble mediated gas transfer is weak for winds up to 30-35 m/s for gases of most solubilities for fresh water and for sea water. We therefore disagree with the assertion that "bubble-related gas transfer is very important for CO2" for winds lower than 30-35 m/s. In addition, Nagel et al. 2015 found no differences between simultaneous heat transfer and gas transfer measurements for wind speeds up to 12.7m/s, indicating that bubble enhancement for gas transfer is not significant at those wind speeds. |
| | Thus we think that the differences found between EC measurements of CO2 and DMS must have causes other than bubbles. |

| | |
|---|---|
| Several studies suggest that interfacial gas transfer appears to be limited at higher winds. This is attributed to wave shielding and other wave-related effects demonstrated in the laboratory by Mueller and Veron (2009) and incorporated into gas transfer models by Fairall et al., 2011 and Donelan and Soloviev, 2016. Such processes could be salient here in relating active thermography to gas transfer. This is not to say that the arguments in the paper about fetch and surfactants etc. are not very well founded. I think they are. | It is true that several studies found a limitation of the Drag coefficient at higher wind speeds (Mueller and Veron 2009, Takagaki et al. 2012). However, the gas transfer velocity has no such limit, see the studies referenced above (Iwano2013, Krall&Jähne2014, McNeil&D'Asaro2006).

Since, as argued above, bubble effects are weak up to 30-35m/s, we can assume that transfer velocities measured at wind speeds below 30-35 m/s are controlled by the transfer through the water surface, and are independent of the gas or measurement technique (EC or DT or ACFT) used. |
| But the overall premise that the dual tracer and active thermography measurements should measure the same thing seems open to debate. I think this should be considered by the authors and perhaps addressed in the manuscript. | Simultaneous measurements of heat transfer and gas transfer with a mass balance method have shown that heat transfer velocities can be scaled to gas transfer velocities for wind speeds up to at least 12.7 m/s (Nagel et al. 2015). No evidence of bubble contribution was found in the measured gas transfer velocities. Therefore, we think that a comparisons with a dual tracer parameterizations is valid. |
| | Changes to the manuscript: we have extended our reasoning why we think that surfactants are the most likely cause for lower gas transfer in the Aranda2010 campaign. We discuss why we think that bubbles do not explain the lower gas transfer using the arguments given above. |
| | references:

Iwano et al. 2013:
Mass transfer velocity across the breaking air–water interface at extremely high wind speeds
https://doi.org/10.3402/tellusb.v65i0.21341

Krall & Jähne 2013:
First laboratory study of air–sea gas exchange at hurricane wind speeds
https://doi.org/10.5194/os-10-257-2014

McNeil & D'Asaro 2006:
Parameterization of air–sea gas fluxes at extreme wind speeds
https://doi.org/10.1016/j.jmarsys.2006.05.013

Mueller & Veron 2009:
Nonlinear Formulation of the Bulk Surface Stress over Breaking Waves: Feedback Mechanisms from Air-flow Separation
https://doi.org/10.1007/s10546-008-9334-6

Takagaki et al. 2012:
Strong correlation between the drag coefficient and the shape of the wind sea spectrum over a broad range of wind speeds
https://doi.org/10.1029/2012GL053988

Nagel et al. 2015:
Comparative heat and gas exchange measurements in the Heidelberg Aeolotron, a large annular wind-wave tank
https://doi.org/10.5194/os-11-111-2015 |

---

## Author Comment (AC2) · 22 Jan 2019

**Authors' response to the 'Interactive comment to "Variability of air-sea gas transfer velocity on the Baltic Sea"' by Anonymous Referee #2**

We would like to express our gratitude to the Anonymous Referee #2 for their thorough and helpful review. Our answers are given in the table below.

| Reviewer's comment | Authors' response |
|---|---|
| 1. The purposes of the manuscript is not clearly formulated. The title suggests that the purpose was to demonstrate how the air-sea gas transfer varies in the Baltic Sea, but inside the text suggests other objectives. Is the objective to compare active thermography method with dual trace method? Or is the purpose to prove that fetch and surface active material have a significant effect on the gas transfer velocity. Or is the purpose of proving that in the transfer velocity study should we use the Schmidt exponent dependently on water conditions? | The purpose is indeed manifold. First, to present the gas transfer velocities we measured in the baltic sea. Second, to stress that under (even suspected) surfactant influence, the correct Schmidt number exponent should be used. And third but not least to interpret the measured gas transfer velocities with respect to fetch/wave age and surfactants.

The dual tracer parameterization by Ho et al. 2011 was chosen to compare our results against, since this study is based on the most thorough compilation of dual tracer data, and it is one of very few studies providing a confidence interval for their parametrization. Since generally dual tracer studies agree quite well with eddy covariance measurements, we could also have chosen a parametrization developed with eddy covariance data, without changing our conclusions.

To better represent the purpose of the manuscript, we changed the title to a more general
'Measurements of air-sea gas transfer velocities in the Baltic Sea' |
| 2. The article in its current form - short, specific information on a given topic, without any extensive descriptions of the transfer velocity - is attractive for scientists who are interested in this topic. But for scientists who are not familiar with this topic, this article may be embarrassing, ........ | The audience, that this article was intended for was indeed researchers, who personally work in the field of air-sea gas exchange or in very closely related fields. It was specifically not intended to be an overview of air-sea gas exchange research or measuring techniques for a general audience. |
| 2. cont. ....... because they will learn specific results from the use of a given measurement technique, but will not know anything about why this is happening. This article is of a purely technical nature rather than a scientific article...... | Schmidt number scaling is done to provide gas transfer velocities, which are explicitly not depending on the measuring technique, and also not depending on the tracer used (different gases, or, in our case. heat). Therefore, the results we presented are not 'results from the use of a given measurement technique', but results any technique would have yielded given the same boundary conditions (ie. the same wind speed, fetch, surfactant coverage..). We added a sentence ('Schmidt number scaling is used to provide a value for the gas transfer velocity, which is independent of the specific measurement technique or tracer.') to paragraph 3.2 to stress that. |
| 2. cont. .... It is important to add more information about the different methods that are used for study transfer velocity or more information about the gas transfer velocity itself and in the Baltic Sea. | Different methods commonly used (dual tracer and eddy covariance techniques) are already mentioned in the introduction. We added a reference in the introduction to an overview paper where measurement methods are described in detail. (' \citet{wanninkhof 2009} gives an overview of the most commonly used techniques to measure the gas transfer velocity.') |

| | |
|---|---|
| a. The Introduction should be extended about information described above. A few sentences about various technique to study gas transfer. What is the ACTF method characterized. Some more information about correlation k with u*. Perhaps more information about variability of air-sea gas transfer velocity in the Baltic Sea, as the title suggested. | Describing the ACFT in the introduction is not necessary, since a brief description of the ACFT is given in section 3.2. We added a reference to section 3.2 to the introduction.

also see answers to point 2. |
| b. More information why mainly wind speed is taken into consideration in air-sea study and why we should add more factors. Not only write the other factors. | added statement 'since wind speed is the most readily available parameter.' to sentence 2 in the introduction.

Some factors influencing the air sea gas transfer velocity other than wind speed are mentioned in the introductory paragraph of section 2. Two selected factors (Surfactants and fetch/wave age) are discussed in detail in sections 2.1 and 2.2. There, experimental evidence is given in the form of references that both factors are indeed modifying the gas transfer velocity and need to be taken into account. |
| c. P2L6 . . .the active controlled flux (CFT). . . | abbreviation added & sentences slightly rearranged |
| d. P2L 9 add references after: A wealth of studies (references) have shown. . .. | We decided to group the 'wealth of studies' using the respective factor each of them addressed, respectively. So all references that follow later in the sentence are selected studies from the aforementioned 'wealth of studies'. |
| e. P3L27..the active controlled flux (CFT) | abbreviation added |
| f. Please exchange para 4.1 with 4.2 for better organization, as you mention in presence para 4.1 cruises which are introduce in presence para 4.2 | we swapped paragraphs 4.1 and 4.2 |
| g. P7L29 - . . .from 25 April 2009 until 7 May 2009 on the German. . .. | replaced 'March' with 'May' |
| h. P9L11 During most of the FS Alkor campaign in 2010
...
.. this should be after Fig. 5 where you introduce this cruise | Figure placement will be taken care of in the production stage of the finalized manuscript.
We added a clarifying statement that this paragraph is about the Alkor 2010 campaign in the previous sentence, where the fig 5 is referenced for the first time. |
| i. The results and conclusion are very short when they are the most important part of the article. Maybe comparison with other data from the Baltic Sea. | Unfortunately, not many measurements of gas transfer in the baltic sea exist. We have added a sentence mentioning both previous measurements of gas transfer in the Baltic, and compare our results with one of them (Fig.6). We have significantly extended the discussion of our results, to more thoroughly argue that surfactants are the most likely reason why the Aranda2010 results from the Archipelago are lower than expected. We have also added a discussion of why we think bubble mediated gas transfer is no valid explanation for this to comply with the requests of the other two reviewers. To do this, we have added one eddy covariance based parametrization of $CO_2$ gas transfer and two eddy covariance based data sets of DMS transfer to Fig. 6. |

| j. We know that at higher and lower wind speed gas transfer are limited so maybe more about that. | The ACFT method cannot be used to measure gas transfer at very low wind speeds. Therefore, discussing limits in the low wind speed case is outside of the scope of the presented manuscript.

For high wind speeds, the gas transfer velocity was found to be not limited, see:

McNeil & D'Asaro 2006:
Parameterization of air–sea gas fluxes at extreme wind speeds
https://doi.org/10.1016/j.jmarsys.2006.05.013

Iwano et al. 2013:
Mass transfer velocity across the breaking air–water interface at extremely high wind speeds
https://doi.org/10.3402/tellusb.v65i0.21341

Krall & Jähne 2013:
First laboratory study of air–sea gas exchange at hurricane wind speeds
https://doi.org/10.5194/os-10-257-2014 |

---

## Author Comment (AC3) · 22 Jan 2019

**Authors' response to the 'reviewer comments for OS-2018-108' by Anonymous Referee #3**

We would like to express our gratitude to the Anonymous Referee #3 for their thorough and helpful review. Our answers are given in the table below.

| Reviewer's comment | Authors' response |
|---|---|
| My only major comment is the heavy focus on comparing with the dual tracer technique. The ACFT approach has a completely different 'time constant' relating k to more or less instantaneous wind with a footprint of a few square meters, whereas the dual tracer technique is quite the opposite. I think a more proper comparison would be with eddy covariance based results. | We have added an eddy covariance based parametrization of $CO_2$ to Fig. 6, as well as two eddy covariance based datasets of the transfer of DMS. We have extended the discussion of the results with respect to those $CO_2$ and DMS measurements. |
| Page 2. Line 12: Waterside convection is also a process which might influence the transfer velocity | We added '[...] and convective mixing (e.g. \citet{rutgersson2011}).' to the end of the first paragraph of Section 2. |
| Page 2, paragraph starting at line 25, the recent paper by Pereira et al. 2018, would also be good to include here. | We replaced the citation Pereira et al. 2016 with the much more recent Pereira et al. 2018, since both use the same technique: gas exchange is measured in a baffle stirred tank with water sampled from the ocean. |
| Page 5, figure 1: how sensitive are your calculated transfer velocities to the variation Sc? | We assume the Schmidt numbers (or Prandtl numbers in the case of heat) to be well known, i.e. as having no uncertainty. The uncertainty of the Schmidt number exponent when Schmidt number scaling is done is included in the uncertainty of the calculated gas transfer velocity (error bars in figs. 3-6). |
| Page 6, lines 25-26: how long averaging period did you use? | The wind speed was averaged for duration of each single measured heat transfer velocity measurement, i.e. about 20 minutes. Weather data was provided by the ships with a temporal resolution of 1 minute (FS Alkor) and 10 seconds (RV Aranda). |
| Page 6, line 27, I think section 4.2 suits better before the current 4.1 section. | We agree. Sections 4.1 and 4.2 are swapped in the updated manuscript. |
| Page 8, figure 3: I would suggest comparing with an EC based parameterization in- stead. Additionally, Ho et al. use a 10-min mean wind speed, what averaging period are you using for your wind speeds? | Concerning the averaging period, see our response to comment no. 5.

We have added an eddy covariance based parametrization for the transfer of $CO_2$ to fig. 6, in addition to two eddy covariance based data sets of the tracer DMS. |
| Page 8-9, line 1: please specify what you mean by "the response time of the system is very high", what is meant by "high" here, how long time is this? | We have reworded and extended this section, to hopefully better describe the relationships between the response time of the water surface and the residence time of a water parcel in the heated patch. |
| Page 9, line 3: similar comment, please specify what the typical response time of the water surface you refer to | Response times can be easily calculated from the heat transfer velocities given in the appendix of the manuscript and Eqn. 4 in the manuscript. For wind speeds of 5m/s and above, they are in the order of 0.3-1.7 s. |

| | |
|---|---|
| Page 9, line 5: again, how long are the residence times estimated from the IR images. | The residence times measured for the conditions below 5m/s wind speed are around 1.6 to 3.3 s. Using Eqn. 4 we can see that we cannot resolve transfer velocities below $k = \sqrt{D_{heat}/\tau_{res}}$. Thus the minimum resolvable transfer velocity is in the order of $k_{heat}$=75 to 100cm/h. However, we do expect lower heat transfer velocities than that at low wind speeds. One can see that by Schmidt number scaling (for instance) the Ho parametrization to a Schmidt number (or Prandtl number) of 7. |
| Page 12, line 10: The Aelotron has already been introduced in the text, no need for a second introduction here. | Those sentences were not intended as another introduction of the Aeolotron, but a justification why we think it is the best wind-wave tank to compare our field data to. We reworded the sentence to better stress what we wanted to say there. |

---

## Referee Report (RR1)

In my review of the earlier submission, I raised the issue of bubble contributions to gas transfer. I am pleased that the authors addressed the issue directly in this revision. Here the authors argue against solubility as an important control for gas transfer effect by invoking laboratory studies in the circular wave tank (Krall, 2013). I agree that would be good evidence against bubbles as a major mechanism of gas transfer in those studies. I wonder if they could offer a little more insight into how the rate of bubble generation in their tank relates to that in the ocean. Is there evidence that wave-breaking occurs at similar rates in the tank as in the open ocean at similar wind stress? If they wish, this could be addressed with minor modification to the text and should not require further review.

Overall, I think the manuscript is a very useful contribution and recommend publication.

A few typos and minor grammatical issues are noted below:

P2 line6 typo "me" should be "be"

P6 line16. The partitioning of the text into paragraphs is a bit random throughout the paper. This was a good example. There's a one sentence paragraph that directly relates to the rest of the text in this section. Why not one paragraph. I noticed this throughout and it was a bit distracting.

P9 Figure 4 caption. Delete comma after "Conditions".

I'm a little confused about Fig 3 and Fig 4. The text says that Figure 4 shows data "in comparison to the same parameterization" used for Alkor 2009. The figure itself shows only 2010 data and there is no comparison shown. Also the wind speed axis scale changed between Fig 3 and 4 which makes it difficult to compare.

P11. Grammatical issues with the caption to Figure 6. This is not a sentence: "For the measurements on RV Aranda in 2010 open ocean conditions, assumed to be with a virtually unlimited fetch, are marked with filled circles, while the fetch limited measurements in the archipelago are marked with open circles."

P12 line 13 spelling

P12 line 15. Another odd one sentence paragraph which could easily be part of the next paragraph.

P12 line 25. Comma is in the wrong place. Should be: "…Aeolotron that, at very…"

P12 line 29 "the the"

P14 line 11 "the the"

---

## Author Response (AR2)

**Author's response to the referee report by Eric Saltzman and the Editor report by Mario Hoppema**

| Reviewer's comment | Authors' response |
|---|---|
| I'm a little confused about Fig 3 and Fig 4. The text says that Figure 4 shows data "in comparison to the same parameterization" used for Alkor 2009. The figure itself shows only 2010 data and there is no comparison shown. | All of the Figs 3, 4 and 5 show the Ho2011 parametrization. That means that Fig 4 shows the same parameterization that Fig 3 (=Alkor2009 data) shows.
However, since this sentence seems to have led to confusion, we have changed it. |
| Also the wind speed axis scale changed between Fig 3 and 4 which makes it difficult to compare. | Axes are now identical for Figs 3, 4 and 5. |
| P2 line6 typo "me" should be "be"
P9 Figure 4 caption. Delete comma after "Conditions".
P12 line 13 spelling
P12 line 25. Comma is in the wrong place. Should be: "...Aeolotron that, at very..."
P12 line 29 "the the"
P14 line 11 "the the" | all corrected |
| **Editor's comment** | **Authors' response** |
| Please change all units into the format with negative exponent, for example m s-1 | changed, also changed in figures. |
| Dates should have a format like 24 July 2015 | done with the exception of Tables A1, A2 and A3, where the full month names would make the tables too wide for a page. The format used in the tables is specified in the table headers. |
| P1, L20 k was explained earlier in the paragraph, but it is not clear what k600 denotes | Added sentence describing what k600 is. |
| P2, L17-18 "A wealth of studies have shown, that, despite the common approach parameterizing the gas transfer velocity to wind speed alone, a multitude of other factors influence gas transfer …" This is a strange construction. Please modify the sentence for better readability. | We split up the sentence into two sentences, hopefully improving readability |
| P2, L26 One factor contributing to the disagreement between gas transfer velocities measured at the same wind speed by different methods are surface … | Canged to: One factor contributing to the disagreement between gas transfer velocities measured at the same wind speed even by the same measurement technique are surface … |
| P5, L12 What does "see 1" mean here? I think it can be deleted. | That was intended as reference to Fig. 1 with the word 'Fig.' missing.
We moved that reference to after '...was used to scale the heat transfer velocities to gas transfer velocities (see Fig. 1).' |
| P12, L23 "young seas" Is this term common? | changed to 'young wave ages' |

| minor grammar issues: | all changed. |
| --- | --- |
| P1, L5 … are much lower than those based on the empirical wind speed … | |
| P1, L18 dimethylsulfide (lower case) | |
| P1, L18 … state of the art methods of measuring … | |
| P3, L5 with a seasonal dependency at a near-shore location (Schmidt and Schneider, 2011). | |
| P5, L4 The references appear better at the end of the sentence. | |
| P5, L7 insert comma after exponent (for more clarity of the sentence) | |
| P6, L17 Schmidt (2) is obviously incorrect, also in the Reference list. Please correct. | |
| P7, L1 … at stations where the vessel … | |
| P8, L1 … by the weather station of each vessel. | |
| P9, L4 … transfer velocities against the wind speed … | |
| P12, L8-10 Because of the high solubility of dimethylsulfide (DMS), this tracer's gas transfer has almost no bubble-induced component; transfer velocities of DMS measured by Bell et al. (2013, 2015) have values very similar to ours, indeed (Fig. 6). | |
| P12, L14 parameterization | |
| P12, L17, 20 m s-1 (format) | |
| P13, L2 probably (not possibly)? | |
| P13, L6 delete Therefore | |
| P13, L9 delete comma after helpful | |
| P13, L31 … these data, some individual measurements … | |

[revised manuscript text omitted]